# The chromatin landscape of high-grade serous ovarian cancer metastasis identifies regulatory drivers in post-chemotherapy residual tumour cells
W. Croft [1,9] ✉, R. Pounds[2,3,9], D. Jeevan[2], K. Singh[3], J. Balega[3], S. Sundar[2,3], A. Williams[4], R. Ganesan[2,4], S. Kehoe[5,10], S. Ott [6], J. Zuo[1,10], J. Yap[2,3,10] & P. Moss [1,7,8,10] ✉

Disease recurrence following chemotherapy is a major clinical challenge in ovarian cancer (OC), but little is known regarding how the tumour epigenome regulates transcriptional programs underpinning chemoresistance. We determine the single cell chromatin accessibility landscape of omental OC metastasis from treatment-naïve and neoadjuvant chemotherapy-treated patients and define the chromatin accessibility profiles of epithelial, fibroblast, myeloid and lymphoid cells. Epithelial tumour cells display open chromatin regions enriched with motifs for the oncogenic transcription factors MEIS and PBX. Post chemotherapy microenvironments show profound tumour heterogeneity and selection for cells with accessible chromatin enriched for TP53, TP63, TWIST1 and resistance-pathway-activating transcription factor binding motifs. An OC chemoresistant tumour subpopulation known to be present prior to treatment, and characterised by stress-associated gene expression, is enriched post chemotherapy. Nuclear receptors RORa, NR2F6 and HNF4G are uncovered as candidate transcriptional drivers of these cells whilst closure of binding sites for E2F2 and E2F4 indicate post-treated tumour having low proliferative capacity. Delineation of the gene regulatory landscape of ovarian cancer cells surviving chemotherapy treatment therefore reveals potential core transcriptional regulators of chemoresistance, suggesting novel therapeutic targets for improving clinical outcome.

High Grade Serous Ovarian Cancer (HGSOC) is the most common and lethal subtype of ovarian cancer, affecting 239,000 women worldwide each year, and has a dismal 5-year survival rate of 25%. The vast majority (80%) of patients are diagnosed with advanced FIGO stage III or IV disease with evidence of metastatic progression[1,2]. Patients often present with widespread dissemination throughout the peritoneal cavity with the most frequent site of metastasis being the omentum[3].

Treatment options for advanced HGSOC include chemotherapy with platinum-based compounds and cytoreductive debulking surgery. Despite initial clinical responses in most cases, chemoresistance is extremely common and 70–95% of patients suffer disease recurrence within 2 years[2]. Mechanisms of chemoresistance have been intensely studied over the past 30 years, largely focussing on resistance-signalling pathway perturbations and gene expression profiles. Mechanisms include dysregulation of platinum compound influx/efflux pumps, DNA damage repair (DDR) pathways and cell death response[4] but the number and complexity of resistance mechanisms to chemotherapy treatment surpass those of more targeted therapies[5].

The somatic mutation profile of HGSOC is complex with extreme copy-number variation, near total prevalence of TP53 mutation[6,7] and

[1]Immunology and Immunotherapy, School of Infection, Inflammation and Immunology, College of Medicine and Health, University of Birmingham, Birmingham, UK. [2]Cancer and Genomic Sciences, School of Medical Sciences, College of Medicine and Health, University of Birmingham, Birmingham, UK. [3]Pan-Birmingham Gynaecological Cancer Centre, City Hospital, Birmingham, UK. [4]Histopathology Department, Birmingham Women's and Children's NHS Foundation Trust, Birmingham, UK. [5]Department of Gynaecological Oncology, Oxford University Hospitals NHS Foundation Trust, Oxford, UK. [6]Warwick Medical School, University of Warwick, Coventry, CV4 7AL, UK. [7]University Hospitals Birmingham NHS Foundation Trust, Queen Elizabeth Hospital, Birmingham, UK. [8]National Institute for Health and Care Research (NIHR) Birmingham Biomedical Research Centre, Birmingham, UK. [9]These authors contributed equally: W. Croft, R. Pounds. [10]These authors jointly supervised this work: S. Kehoe, J. Zuo, J. Yap, P. Moss. ✉e-mail: w.d.croft@bham.ac.uk; p.moss@bham.ac.uk

exceptionally high intratumor heterogeneity. Detailed transcriptional characterisation in metastatic ovarian cancer has identified cell type contextures of tumour, stromal and infiltrating immune populations[8–10]. Chemotherapy has been shown to modulate the transcriptional programs within tumour microenvironments[11–14] and an IL6-associated inflammatory network and immediate early stress response genes have been associated with poor treatment response and accelerated disease recurrence[12]. Furthermore, a recent longitudinal single cell transcriptome study identified and validated a stress-associated tumour cell expression profile to be associated with relapse and reduced progression-free survival[10].

Less well understood are the gene regulatory mechanisms orchestrating transcriptional programs for stress, DNA damage response or cell death signalling that confer chemoresistance. Chromatin accessibility is an important gene regulatory control mechanism and chromatin modifying histone methyltransferase and deacetylase enzymes that modulate chromatin compactness are often altered or mutated in ovarian cancer[15]. Indeed, epigenetic therapies delivered either as single agents or in combination with chemotherapy present a potential option to combat resistance and are now under clinical evaluation[16].

Multi-omic molecular landscape studies have focussed on ovarian samples from normal[17] and primary HGSOC tissue[6,18] but have not studied omentum, which is the dominant site of metastasis. Furthermore, features were not examined in relation to disease stage and as such the clinical importance of chemotherapy-induced modulation of epigenetic profile in the metastatic microenvironment remains undetermined. The omentum also serves as a clinical marker for measuring disease response to chemotherapy[19] which allows comparison of the epigenetic landscape according to chemotherapy response.

Single-cell ATAC-seq measurement of chromatin accessibility is a powerful approach for assessing cellular regulatory features in relation to treatment status. Here we present the first scATAC-seq comparison of HGSOC metastatic omental tumour from treatment-naïve patients compared to tumour following neoadjuvant chemotherapy (NACT). We characterise the chromatin accessibility landscape and transcription factor binding motif enrichments of heterogeneous tumour cells in the metastatic microenvironment and identify shifts in key regulatory patterns following NACT.

## Results
### Chromatin accessibility profiling identifies major lineage cell type composition and tumour heterogeneity within omental metastasis

Tissue from HGSOC omental metastatic deposits were profiled for single cell chromatin accessibility by scATAC sequencing. In total, ten omental tissue samples were collected from nine HGSOC patients (A-I), consisting of five treatment-naïve samples and five biopsies taken following NACT at the time of interval debulking surgery, performed 3–4 weeks after the last cycle of NACT (Fig. 1A). Samples pre_A and post_A were matched pre and post chemotherapy samples from the same patient. After filtering, the average number of cells analysed per sample was 2709 (range 568–7254) (Table S1). Quality checks on these data and read density profiles at the ubiquitously open housekeeping gene TBP genomic loci confirmed consistent transcription start site (TSS) peak enrichment profiles with strong peak signals (Figs. S1 and S2). Independent unsupervised clustering identified a mean of 7.9 clusters per sample (range 4–11) (Fig. S3).

Unsupervised clustering on all-sample integrated data from a total of 27,089 single cells identified 7 clusters (Fig. 1B). Gene activity scores, predicted from chromatin accessibility profile at cell lineage marker gene loci, assigned clusters into 4 major lineage cell types in the treatment-naïve (pre) and post-chemotherapy (post) tumour microenvironment (Figs. 1C and S4A). Gene activity (Fig. 1D), copy number alterations (CNAs) (Fig. 1E) and aggregated chromatin accessibility at gene loci (Fig. 1F) of canonical cell type marker genes of interest were consistent with cell type annotation. The epithelial population was marked by EPCAM gene activity whilst expression of the US FDA-approved HGSOC biomarkers MUC16 and WFDC2[20,21] and HGSOC-specific LAPTM4B[18] confirmed tumour origin. Epithelial tumour

cells had markedly increased DNA CNAs compared to non-malignant cells and copy number loss and gain profiles displayed inter-tumoral differences. Fibroblast, lymphocyte and myeloid populations were identifiable with PDPN/COL1A2, IKZF1/PTPRC and CD14/TREM1 gene activity/accessibility profiles respectively. Inter-tumor heterogeneity of the chromatin landscape was high, with notable patient-specific clusters within the epithelial population (Figs. 1B and S4B, C). Within-sample cell type contextures determined epithelial cells to be the dominant cell lineage in the omental metastasis tumour microenvironment. Whilst all patient samples contribute to the epithelial cell pool, treatment-naïve cells dominate providing 73% of the pool (Fig. 1G and Table S1). Samples with increased epithelial proportion had concurrently reduced myeloid and fibroblast fraction and trended towards shorter survival times (Fig. S5).

### Binding motifs for ovarian cancer oncogenes MEIS and PBX are enriched in accessible chromatin sites of omental metastasis epithelial cells

Regions of accessible chromatin (peaks) were identified that were markers for major lineage cell type and were consistent between samples (Fig. 2A). Motif-level activity was next calculated with chromVar[22] and differential analysis applied to discern the transcription factor (TF) DNA-binding motifs that were differentially enriched within these cell-type defining peaks. This highlights the key transcriptional regulators influencing cell-type specific gene activity within the tumour microenvironment (Fig. 2B). Epithelial cell peaks were differentially enriched for MEIS binding motifs, a TF family that has been shown to be upregulated in HGSOC tumours and associated with growth, invasion, stemness, epithelial-mesenchymal transition, chemoresistance and poor prognosis[23]. PBX2 forms DNA-binding complexes with MEIS and motifs for PBX2 also displayed epithelial cell enrichment. Of note, MEIS and PBX have been previously associated with promoting chemotherapy resistance in ovarian cancer (OC)[24]. Furthermore, binding motifs for the OC stem and platinum tolerant tumour cell marker TP63[25] are substantially enriched, suggesting ongoing activity within TP63-driven gene programs.

Motifs enriched within fibroblast-defining peaks included ATOH7, FOS, TCF7L2 and TCF7, the latter two TFs having key roles in the Wnt signalling pathway[26]. Tumour infiltrating lymphocyte and myeloid populations were enriched for motifs including ETS family transcription factors (ETV5/6 and EHF) and MEF2B, as well as NFKB1 which is a central activator of pro-inflammatory genes.

### The chromatin landscape of residual tumour cells following chemotherapy display increased accessibility enriched for stemness-driving motifs

We next sought to identify how the chromatin landscape of major lineage cell types is modulated following chemotherapy treatment (Fig. 3).

Of note, the relative proportion of cell lineages was broadly stable post chemotherapy with only a trend towards modest reduction in the lymphoid pool (Fig. 3A). However, the chromatin accessibility landscape was substantially different (Fig. 3B, C) and this was most evident within epithelial cells which showed 1428 differentially accessible chromatin sites (DACs) between pre- and post-treatment tissue. In contrast, this value was 110, 152 and 146 for fibroblast, lymphocyte and myeloid cells respectively. Chromatin regions showed substantially more accessibility following chemotherapy, with 1268 sites being 'open' post treatment compared to 160 sites 'closed' in epithelial cells (Fig. 3C). This is noteworthy given the consensus that stem cells and poorly differentiated cancer cells harbour more open 'poised' chromatin relative to well-differentiated cells[27].

Some of the most divergent DACs were notable for peaks representing chromatin that was closed in treatment-naïve samples but open following chemotherapy, including those at the TPK1, FAM135B and TRAPPC9 gene loci. TPK1 is of particular note as it catalyses the conversion of thiamine to thiamine-pyrophosphate and upregulates thiamine metabolism to promote tumour cell progression during hypoxic stress[28]. A smaller number of DACs, such as those at the Ryanodine receptor RYR1 loci, become closed following treatment (Fig. 3D).

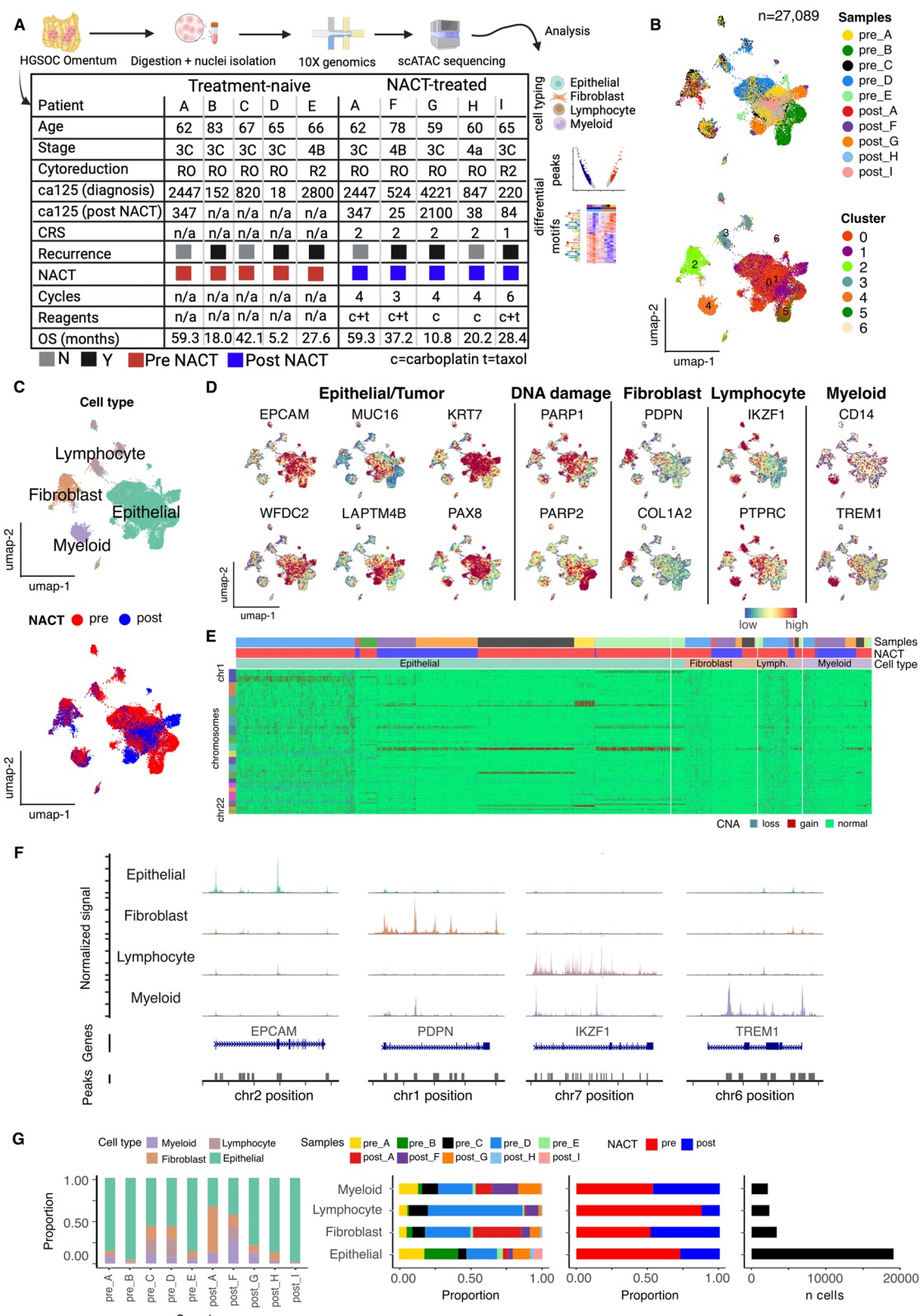

**Fig. 1 | Chromatin landscape defined major cell types in HGSOC omental metastasis. A** Methods and cohort summary. CRS chemotherapy response score (1 = poor; 2 = moderate); Cytoreduction (R0 = removed all macroscopically visible disease; R2 = disease of min 1 cm remains following surgery); Blood CA-125 level (u/ml). Created with BioRender. **B** UMAP embedding of all patient-integrated scATAC data overlaid with patient label (top) and unsupervised clustering label (bottom), **C** UMAP embeddings overlaid with major cell lineage annotation and neo-adjuvant chemotherapy treatment (NACT). **D** Gene activity scores of major cell type defining genes. **E** Profile of copy number alterations (CNAs) detected across all cell types. **F** Major cell lineage aggregated chromatin accessibility profile at major cell lineage defining gene loci. **G** (left-to-right) Cluster composition of each sample; Sample composition, NACT treatment composition and total number of cells for each major lineage cell type.

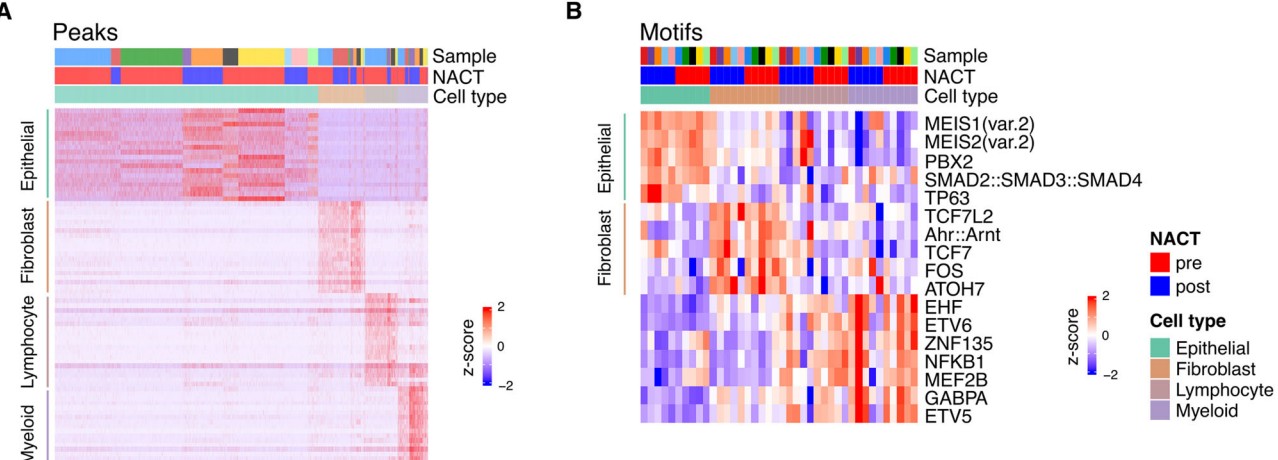

**Fig. 2 | Chromatin accessibility profile: peaks and enriched motif markers of major cell types in HGSOC omental metastasis. A** Single-cell chromatin accessibility profile at major lineage defining marker peaks (shown are top 20 differential peaks by cell type having adjusted $p < 0.001$). **B** Patient-cell-type average chromVar transcription factor motif activity score profile of differentially enriched cell type marker motifs (shown are top 10 motifs by cell type having adjusted $p < 0.001$).

Differential motif level analysis was next used to identify transcription factor binding motifs that were differentially enriched in treatment-naïve vs. post chemotherapy epithelial tumour cell accessible chromatin loci (Fig. 3E), aiming to identify transcription-factor-driven gene expression programs most affected by chemotherapy. Binding motifs for the developmental regulator and emerging cancer gene MEF2 family TFs[29] are enriched in treatment-naïve epithelial cells. Post chemotherapy, the binding motifs of many transcription factors with strong links to stemness and platinum resistance are differentially enriched in open chromatin sites. These include the tumour suppressor protein TP53, oncogenic factor TP63 and the Wnt signalling activator TCF7. The TP53 protein (p53) is almost ubiquitously mutated in HGSOC and all tumour samples regardless of treatment status had consistent TP53 mutations with aberrant p53 expression (Fig. S10). The divergence of motif enrichment was much less distinctive for fibroblasts and infiltrating immune cells. Despite this, some motifs of note include the angiogenesis and EMT-linked TF FOXC2[30] enriched in post NACT fibroblasts as well as ZBTB7B (ThPOK) showing enrichment in treatment-naïve myeloid cells (Fig. S6).

To home in on the likely 'active' master tumour cell regulators, we identified the transcription factors that, along with binding motif enrichments, also had concurrent increased gene activity in post/pre-treatment cells (Fig. 3F). This highlighted 12 candidate transcriptional regulators in pre-treated metastatic cells which included MEF2B, KLF12 and STAT2. In the tumour cells that reside post treatment, 40 candidates were found to have increased expression alongside enriched accessible binding motifs, including TP63, TCF4 and the well-defined regulator of OC cancer stem-cell differentiation, TWIST1[31]. Expression of these TFs in tumour cells is supported by accessibility in the gene body/promoter (Fig. 3G) and they represent candidate important factors shaping the post-chemotherapy transcriptional landscape of tumour cells.

To assess the relevance of such divergent regulation of transcriptional programs on tumour cell functionality following chemotherapy, we assessed the gene activity scores of selected hallmark gene modules of interest (Fig. 4). Residual tumour cells post NACT were enhanced for several relevant pathways including DNA damage, Apoptosis, IL6-Jak-Stat3, Hedgehog, Wnt beta catenin signalling, and inflammatory response whilst a mitosis gene set was reduced (Fig. 4). Interestingly, bile acid metabolism was also increased, with bile-acids thought to play largely protective roles and implicated as a novel modality in the chemotherapy of ovarian cancer[32].

**Transcriptional drivers of stress-associated chemoresistance are identified in tumour cells**

The chromatin landscape specifically of 19,122 epithelial tumour cells was next examined in further detail (Fig. 5). UMAP embeddings overlaid with sample label and unsupervised clustering (Fig. 5A), HGSOC tumour and DNA damage marker gene activity, and hierarchical clustering on CNAs (Fig. S7) are suggestive of a high degree of intra and inter-tumoral heterogeneity. Profiling clustered sub-populations highlights heterogeneity of gene, peak and motif activity within tumour sub-populations (Fig. S8) All samples had prominent EPCAM, LAPTM4B, KRT7, PAX8 gene activity plus MEIS1 motif enrichment, supporting epithelial cell type, but other tumour markers including MUC16 showed more sample-wise variation (Fig. 5B). Selected motifs of interest identified as having differential enrichment following chemotherapy included MEF2 family, TP53 and TP63. In all but one of the treatment-naïve samples, MEF2A/B enrichment was observable in >60% of the pre-treated cells whilst TP53/63 enrichment was prominent in 40–60% of the post-treatment populations (Fig. 5B).

HGSOC chemoresistance within omental metastasis has been shown to be promoted by a defined stress-associated gene signature (Table S2) detectable in a subset of pre-treated tumour cells[10] and as such it was important to examine this tumour-stress gene activity score in these data (Fig. 5A). In concurrence with previous findings, both the per-cell and per-sample mean tumour stress scores were significantly increased following chemotherapy, indicating possible selection for chemoresistant stress-associated tumour cells (Fig. 5B, C). Of note, the two patients in the cohort who suffered disease recurrence (post_F and post_G) were also those with the highest tumour stress scores.

To investigate master transcriptional regulators driving this stressed/chemoresistant tumour cell state, we identified per-sample chromVar motif score correlates with stress signature score (Fig. 5D). This identified binding motif activity that was significantly correlated with stress score and included the transcription factors RORA, GATA, NR2F6 and HNF4G. Interestingly, motifs with the strongest inverse correlation included E2F2 and E2F4, which are activating regulators of DNA replication and cell cycle. These imply chemoresistant stress-associated tumour cells to have low proliferative rate and aligns with prior evidence of enhanced chemotherapy escape in slow-cycling tumour cells[33].

**Chromatin landscape of tumour sub-populations in a pre/post NACT matched case with long survival outcome**

Intriguingly, the patient with substantially lowest post-chemotherapy tumour stress score (patient A) had the longest overall survival with no evidence of disease recurrence at just under 5 years following diagnosis. As such, we next directly compared the chromatin landscape in epithelial tumour cells before and after neo-adjuvant chemotherapy from this individual (Fig. 6).

In total, 4063 tumour cells were studied of which 3365 were from pre-treatment samples (83%) whilst 698 cells (17%) were from post-treated

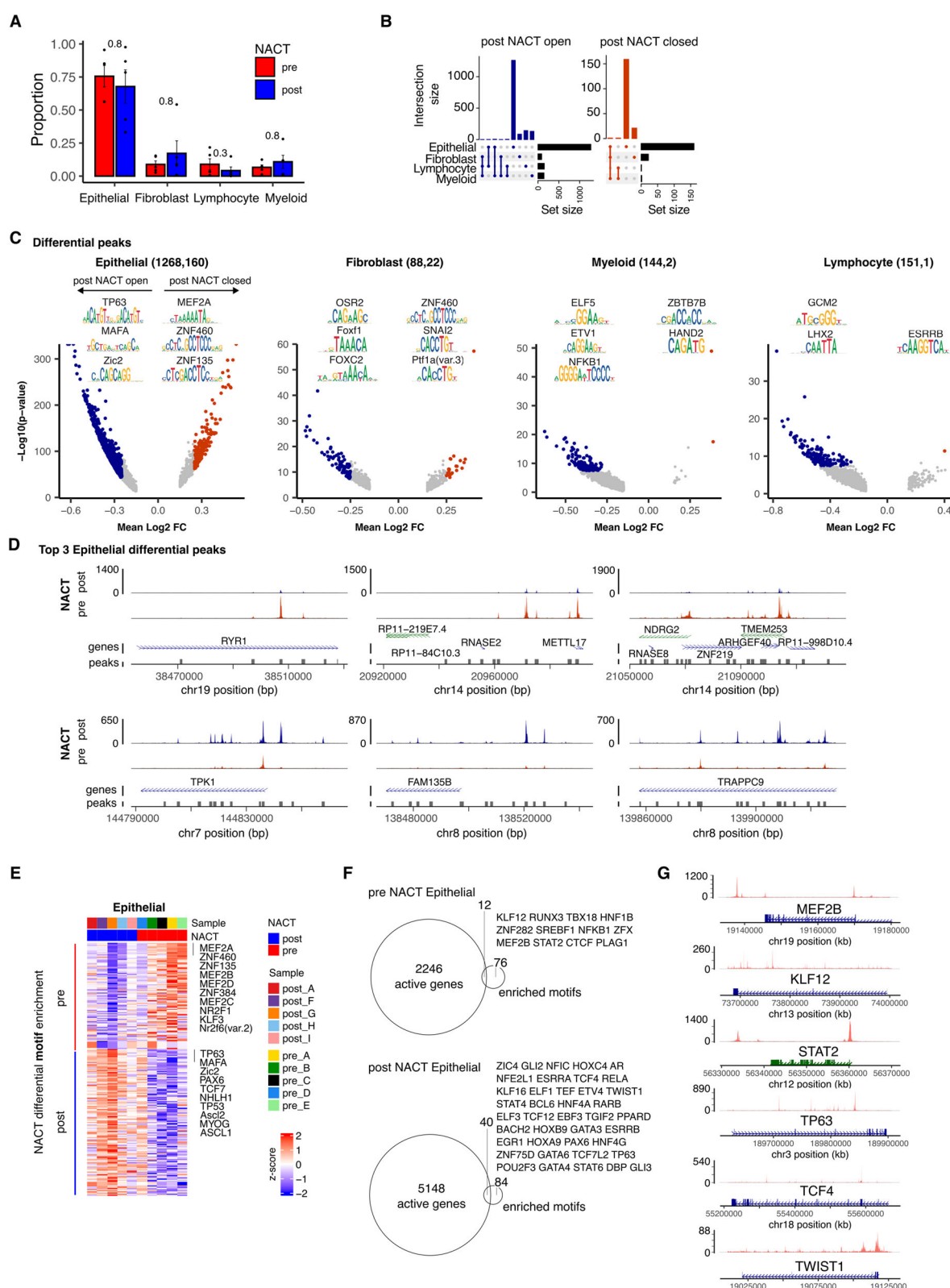

**Fig. 3 | Modulation of the chromatin accessibility landscape and transcription factor binding motif enrichments following chemotherapy. A** Proportion of total stratified by chemotherapy. Points represent the within-sample cluster fraction and *p* value determined by Mann–Whitney test. **B** Summary counts of differentially accessible chromatin (DAC) sites open/closed post chemotherapy. **C** Differential accessibility of chromatin sites between treatment-naïve (pre) and post chemotherapy (post) samples. Coloured points indicate DACs (adjusted *p* < 0.01 and absolute log2FC > 0.25). Motif sequence logos presented for the top 3 motifs enriched within open/closed peak sites. **D** Aggregated accessibility profile of selected DACs stratified by treatment. Tracks Y scale represents normalised fragment count. **E** Within patient-cell-type average transcription factor motif chromVar activity score profile of motifs identified as differentially enriched (adjusted *p* < 0.01). **F** Intersections of differentially increased gene activity (expression) with enriched motifs in pre NACT treated tumour cells (top) and post NACT-treated tumour cells (bottom). **G** Aggregated accessibility profile of selected transcription factors.

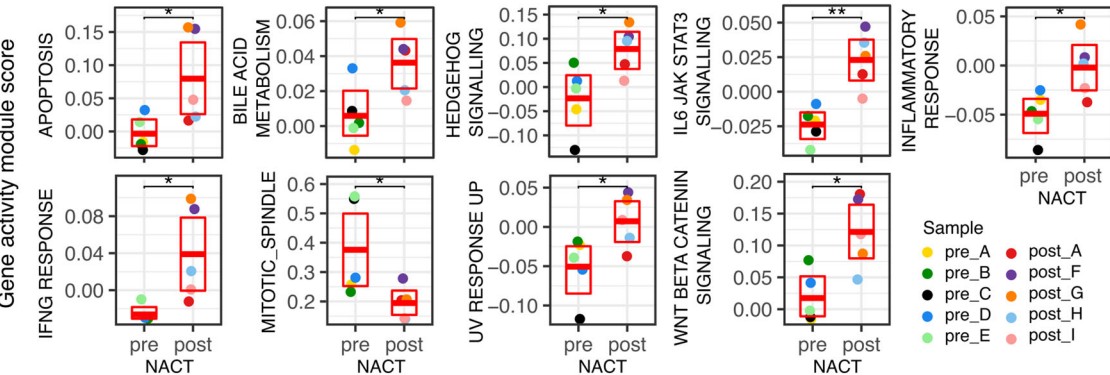

**Fig. 4 | Epithelial cell hallmark gene activity module scores following chemotherapy.** Mean module scores for selected MSigDB Hallmark gene sets calculated on epithelial tumour cells for each sample and stratified by neoadjuvant chemotherapy treatment. Wilcoxon rank sum test **p < 0.01, *p < 0.05.

tissue (Table S1), indicating broadly successful clearance of the initial tumour population (Fig. 6A). Reduced tumour burden was also supported by blood ca125 measures which were 86% reduced post-chemotherapy (Fig. 1A). Unsupervised clustering of chromatin accessibility profiles identified 5 tumour sub-clusters in both pre and post treatment populations (Fig. 5A). Of note, the proportion of cells in cluster 1 increased markedly from 18% pre-NACT to 50% after treatment (Fig. 6B).

Differential analysis of gene activity and motif enrichment identified 5114 genes and 286 motifs to be divergent between pre- and post-NACT tumour cells (Fig. 6C). CNA profiles identified a range of genetic lesions in pre-treatment tumour cells, most notably gains in chromosomes 3, 19 and 20, whilst fewer CNAs were seen in tumour cells following chemotherapy (Fig. 6D). To highlight the likely transcriptional regulators marking the largely chemosensitive (pre-treatment) and chemoresistant (post-treatment) tumour cells, we next determined the intersection of genes and TF motifs that had differential activity following chemotherapy (Fig. 6E). This allowed identification of TFs with concurrent high gene activity plus enriched binding motif in chemosensitive and chemoresistant cells respectively. KLF15, MEF2B, NRF1 and CUX2 were identified as four candidate master regulators of chemosensitivity whilst an additional 37 transcription factors were highlighted as having potential importance in chemoresistance (Fig. 6E).

## Discussion

The relative resistance to chemotherapy in sub-populations of HGSOC tumour cells leads to high rates of disease relapse and is arguably the greatest current challenge in ovarian cancer. Metastatic omental tumours are present in the majority of advanced HGSOC cases[3] and associate with poor clinical outcomes. Our scATAC-seq analysis of omental metastatic tissue provides a range of insights into the fundamental biology and therapeutic targeting of this cancer of unmet need.

Tumour cells displayed a heterogeneous profile of chromatin accessibility, a finding compatible with prior observations and interrogation of scRNA-seq datasets[8,34]. This is likely to be a key facilitator of drug resistance, and subsequent tumour progression, and may underpin the high rate of clinical recurrence and metastasis in ovarian cancer[35]. A range of open chromatin genomic loci were identified as markers of epithelial tumour cells, depicting regions of the tumour genome that are accessible to transcriptional regulation by master transcription factors (MTFs). MTFs can be subverted to control oncogenic transcriptional programs during tumorigenesis and many MTFs have been predicted from pan-cancer expression datasets[36]. ATAC-seq allows interrogation of binding site motif enrichment within open chromatin DNA regions, thus allowing identification of master regulators with high confidence. MEIS homeobox proteins have been associated with tumorigenesis, metastasis and invasion in ovarian cancer[23] and MEIS1/MEIS2 transcription factors were significantly enriched in accessible tumour regions. MEIS forms complexes with PBX, another transcription factor commonly dysregulated in cancer[37], and PBX2 motifs were also enriched, substantiating a primary role for MEIS-PBX in the

transcriptional regulation of HGSOC. The central importance of MEIS-PBX is further indicated by high levels of mRNA and protein content in ovarian cancer[24] whilst overexpression of PBX can lead to platinum-based chemotherapy resistance[38]. These findings extend understanding of the mechanistic role of MEIS1/2 and PBX2 in the evolution of HGSOC metastasis and warrants further investigation.

Comparison of the chromatin accessibility landscape in treatment-naïve and post-chemotherapy tumours identified multiple genomic loci as differentially accessible in epithelial tumour cells. Strikingly, the chromatin was comparably more open in tumour cells that survived initial chemotherapy treatment. Stem cells and early differentiated cancer cells have more open 'poised' chromatin[27] and chemotherapy may provide a selective pressure that favours such less-differentiated cells. Of particular interest were several open regions at the *TPK1* gene locus, spanning both the gene and regions upstream of the promotor. Tumour cells adaptively increase vitamin B1/thiamine intake during hypoxic stress and in response to chemotherapeutic agents and TPK1 is a key enzyme facilitating rapid thiamine metabolism[28]. TPK1 has itself been shown to modulate drug and radio-sensitivity of cancer cells[39,40] and therefore presents a plausible target in the context of ovarian cancer chemoresistance.

Motif-level analysis identified enrichment for the MEF2 family transcription factor binding motifs MEF2A/B/C/D in treatment-naïve tumour cells, implicating these TFs as major drivers of ovarian tumour transcriptional programs. MEF2 promotes survival in a range of cell types and abnormal regulation has been linked to oncogenicity in several tumours[29]. MEF2D is known to be overexpressed in ovarian cancer and is believed to contribute directly to chemotherapeutic resistance[41]. Of note, class IIa histone deacetylases can bind MEF2 and modulate its activity from a transcriptional activator to a repressor. Indeed, HDAC inhibitors have emerged as new therapies to target platinum resistant epithelial ovarian cancer[42] and this may in part be acting via the modulation of MEF2 transcriptional programs.

Following chemotherapy, tumour cells shifted towards regions of open chromatin at binding sites for transcriptional regulators of drug resistance, stemness and Wnt signalling. Two of the most intriguing sites in this regard that also showed concurrent increased gene activity were the oncogenic factor TP63 and stemness regulator TWIST1. TP63, a homologue of the ubiquitously mutated TP53, is expressed in epithelial ovarian tumours[43], has key roles in embryonic development, stemness and suppression of tumorigenesis/metastasis, and can interact with mutated TP53[44]. Via activation of the Wnt signalling receptor, subsequent TP63 activation has been identified to upregulate glutathione pathways to protect against chemotherapy-induced oxidative stress[25]. Nutlin-3a, which inhibits the interaction between mdm2 and TP53, has shown activity against TP63 and may be of interest for future assessment in HGSOC[45]. TWIST1 has been implicated in stemness and chemoresistance in many other cancer types[46], drives cisplatin resistance in an OC model[47] and induces degradation of β-catenin during the differentiation of OC stem-like cells[31]. Our data points to TWIST1 as a key master regulator in metastatic HGSOC warranting further exploration.

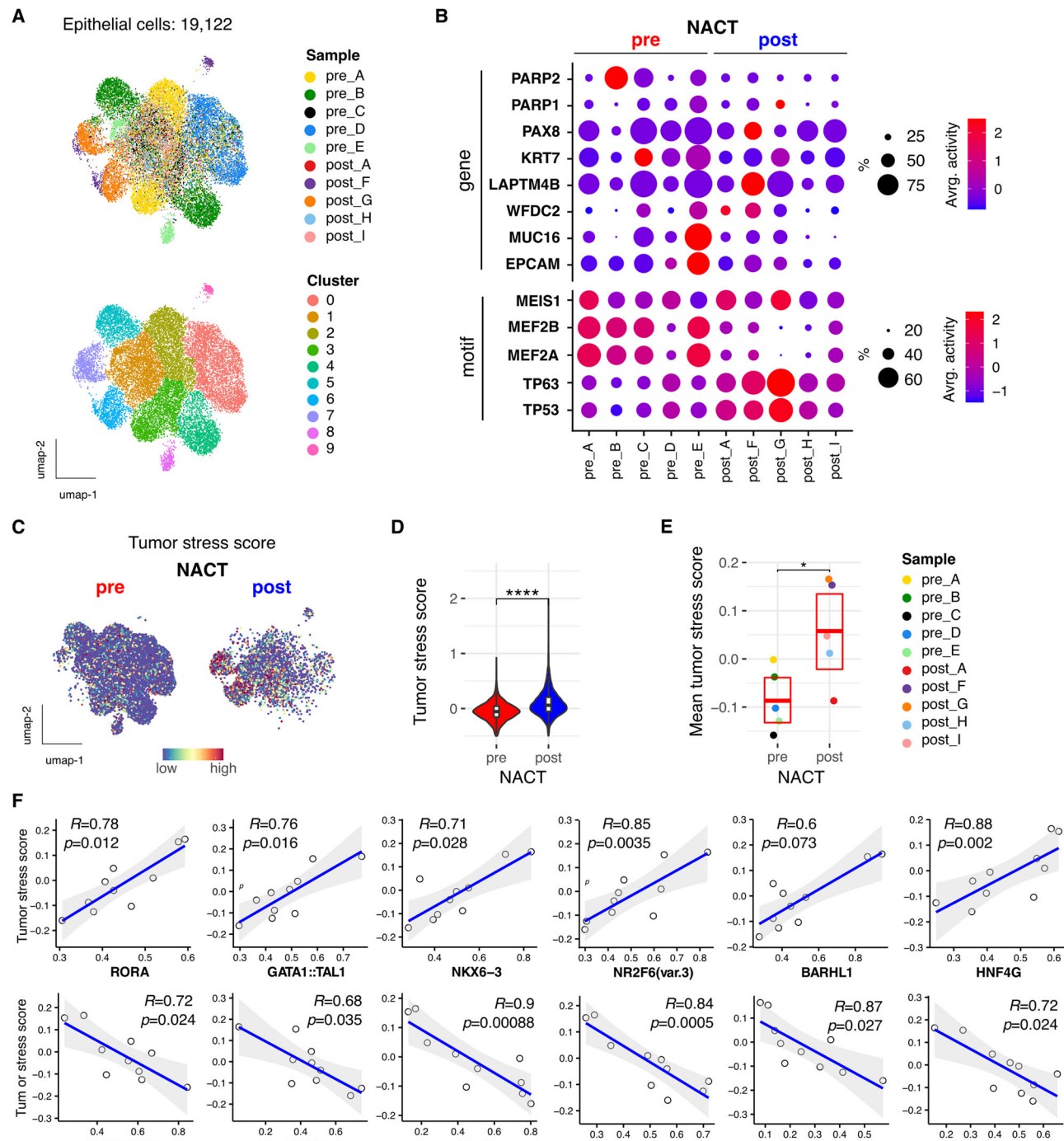

**Fig. 5 | Transcriptional drivers of chemotherapy enriched stress-associated tumour cells. A** UMAP embedding of epithelial cells overlaid with sample label (upper) and cluster (lower). **B** Dot plot gene and motif activity profile of selected tumour genes/motifs of interest. Dot size indicates the percentage of the population showing gene/motif activity. **C** UMAP embedding of epithelial cells stratified by pre/post chemotherapy treatment overlaid with tumour stress signature gene activity score. **D** Distribution of per-cell stress score stratified by chemotherapy. **E** Distribution of per-sample mean stress score stratified by chemotherapy. **F** Significant Spearman rank correlation of per sample chromVar motif activity score with tumour stress signature gene activity score (top 6 significant +ve and −ve correlations are shown).

Wnt/β-catenin signalling has a critical role in driving ovarian cancer chemoresistance[48] and binding motifs for TCF7, one of its downstream transcriptional targets, were also enriched within tumour after chemotherapy. Wnt signalling pathways have been intensely studied due to their role in promoting cancer stemness and drug resistance[49,50], and the potential co-occurrence of pathway activation with enhanced chromatin accessibility of motifs for downstream signalling targets TCF7 and TP63 could be an important modality to explore.

Ovarian tumour cells with stress-associated transcriptional response have also been shown to expand following chemotherapy. We corroborated these findings and also identified the likely transcriptional drivers. Of note, cell cycle promoting TFs, including E2F2 and E2F4, showed decreased enrichment with increasing stress score highlighting low proliferative turnover in chemoresistant cells, a factor that may contribute towards relative resistance to cisplatin-based chemotherapy. Three TFs with strong positive correlations with stress response were RORa, HNF4G and NR2F6,

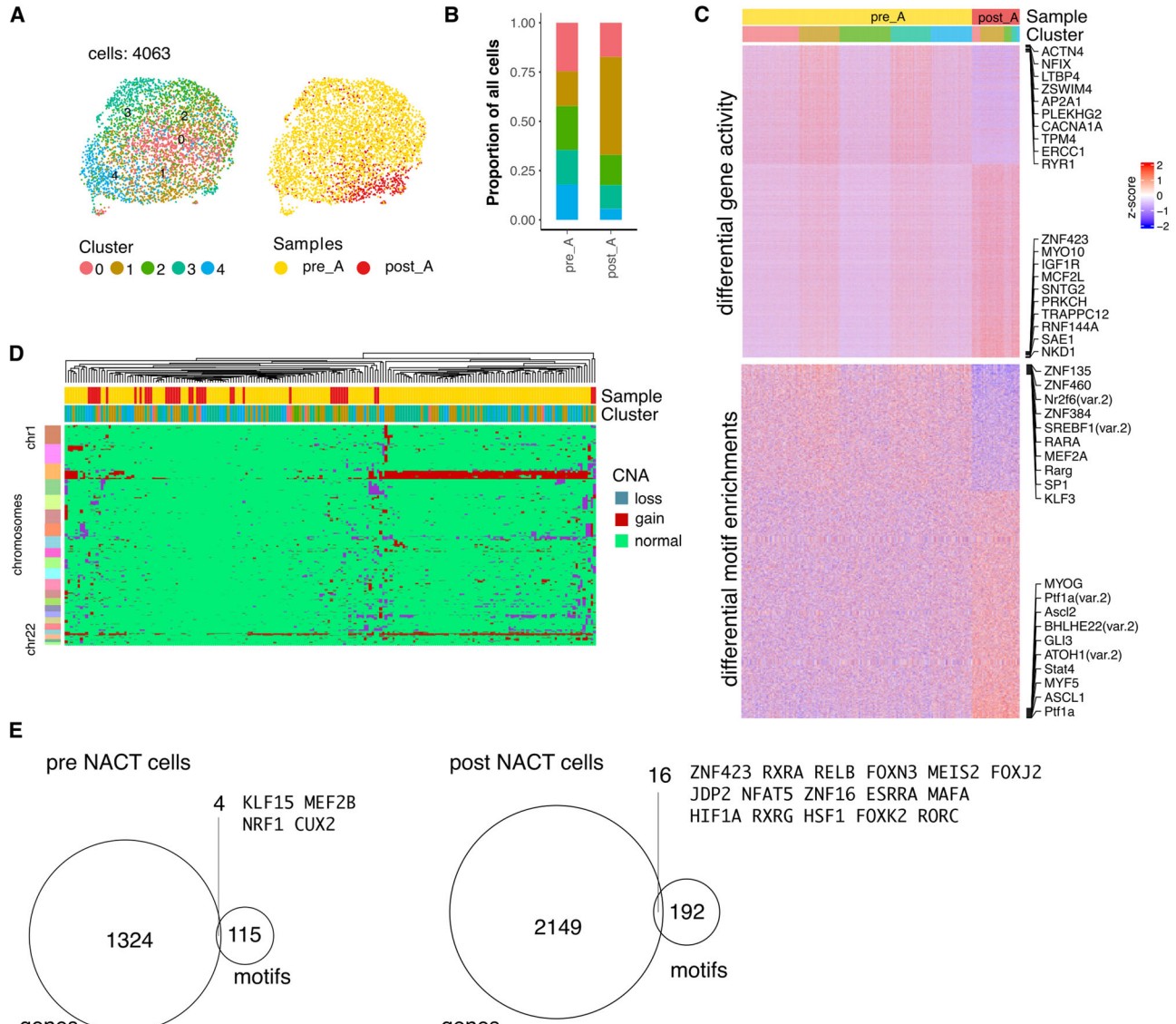

**Fig. 6 | Matched pre vs. post chemotherapy tumour cell chromatin profile in a patient with comparably long progression free survival. A** UMAP embedding of all epithelial tumour cells from patient A overlaid with unsupervised clustering and pre/post chemotherapy sample label. **B** Cluster proportions stratified by chemotherapy treatment. **C** Differentially active genes and differentially enriched transcription factor binding motifs between pre vs. post chemotherapy-treated tumour cells. Differential genes (adjusted $p$ value < 0.001 and absolute log2FC > 1); differential motifs (adjusted $p$ value < 0.001); Top 10 significant (ranked by fold change) up/downregulated are labelled. **D** Hierarchical clustering of copy number alteration profile. **E** Intersections of differentially increased genes with differentially enriched motifs present in pre NACT treated tumour cells (left) and post NACT-treated tumour cells (right).

all members of the nuclear receptor (NR) superfamily. Nuclear receptors are master regulators of tumorigenic processes and are considered as highly druggable cancer targets[51]. These included the retinoid orphan nuclear receptor alpha (RORa), a critical regulator of malignant phenotype. RORa can inhibit tumour cell proliferation and epithelial-mesenchymal transition (EMT)[52] and has been identified as a negative regulator of the Wnt/β-catenin pathway and potential tumour suppressor in multiple cancers[53,54]. Furthermore, NR2F6 expression has been identified to confer cisplatin resistance in epithelial ovarian cancer cells[55]. These data indicate that chromatin accessibility limiting tumour cell proliferation may be important in enabling tumour sub-populations to escape chemotherapy killing.

Focussing solely on matched pre- and post-chemotherapy samples offered insight into the impact of NACT on the gene regulatory landscape of tumour cells in a patient with long-term disease-free survival. In addition to changes in chromatin profile it was also notable that genomic copy number alterations were less frequent following chemotherapy, perhaps indicating residual tumour as less differentiated cells having accumulated fewer

genomic lesions in this case. It was notable that retinoic acid (RA) signalling TFs, including RA nuclear receptors RXRA, RXRG, and RXR-interacting partner ZNF423[56], all had increased activity in these residual tumour cells.

A limitation of our study due to tissue availability meant that pre vs. post NACT comparisons were largely unpaired with one matched sample pair included in the analysis. Contribution of inter-patient tumour heterogeneity makes it difficult to determine if all/some of the residual tumour cells were pre-existing and inherently chemoresistant prior to treatment, or if resistance emerges in some cells following chemotherapy challenge. Data presented here highlights multiple transcription factors with high likelihood to be of importance in shaping tumour cell resistance but further work to functionally evaluate these factors is required.

In conclusion, analysis of the chromatin landscape of metastatic ovarian tumour cells reveals considerable heterogeneity with accessible chromatin enrichment of binding motifs for the TF cancer oncogenes MEIS and PBX. The post-chemotherapy tumour cells have increased chromatin accessibility, including regions accessible to regulation by stemness and

resistance-driving factors such as TP63, TP53, TCF7 and TWIST1. Residual epithelial cells also display enhanced tumour stress signatures with associated candidate TF regulation. These data provide insights into the gene regulatory mechanisms of transcriptional programs in metastatic ovarian cancer following chemotherapy and identify a range of potential therapeutic targets warranting further exploration to combat chemoresistance and improve outcomes in HGSOC patients.

## Methods

### Ethical statement
Participants provided written informed consent to participate in the study. The study was approved by IRAS ID: 225991 and was conducted in accordance with the local legislation and institutional requirements. All ethical regulations relevant to human research participants were followed.

### Tissue processing
Tumour biopsies were obtained following the surgical resection of omental tissue. Macroscopic epithelial tumour cell rich tissue regions were isolated and immediately snap frozen in liquid nitrogen. Biopsies were examined by an Ovarian Cancer expert histopathologist to confirm diagnosis of metastatic HGSOC. Expert also confirmed the presence of neoplastic structures with no evidence of other non-neoplastic epithelial structures within the samples (Fig. S9). All tumours were confirmed as harbouring p53 mutation and advanced stage (FIGO stage III and IV). All patients were negative for BRCA mutation. p53 expression was aberrant overexpression for all tumours apart from patient D who showed a p53 null phenotype (Fig. S10). For NACT-treated patients, treatment consisted of carboplatin or carboplatin + taxol with 3–6 cycles at normal dosage and tissue samples were taken at delayed debulking surgery approximately 4 weeks post treatment. Patient A also received maintenance treatment with the parp inhibitor Rucaparib. Sample preparation involved extracting nuclei from the snap frozen tumour biopsies to generate a single-nuclei solution. A 1X lysis buffer was prepared using 433 µl nuclease-free water (Thermo Fisher Scientific; AM9932), 5 µl Tris-HCl (pH 7.4; Sigma-Aldrich; 77-86-1), 1 µl NaCl (Sigma-Aldrich; S8776), 1.5 µl MgCl2 (Sigma-Aldrich; 63069), 5 µl of Tween-20 (Sigma-Aldrich; P1379), 5 µl Nonidet P40 substitute (Sigma-Aldrich; 74385), 1 µl of Digitonin (Thermo Fisher Scientific; 407560050) and 50 µl of BSA (Miltenyi Biotec; 10-091-376). This was added to 4.5 ml of lysis dilution buffer, consisting of 3.98 ml nuclease-free water, 45 µl of Tris-HCl (pH 7.4), 9 µl of NaCl, 13.5 µl of MgCl2 and 450 µl of BSA, to give a 0.1X lysis buffer solution.

Each tissue specimen was minced using a scalpel into 1–2 mm sections. These were transferred with 1 ml of the 0.1X lysis buffer solution into a 7 ml Dounce tissue homogeniser on ice. The tissue was pushed to the base of the homogeniser and compressed with the larger pestle A for 50 strokes. This was then repeated with pestle B and the sample was transferred into a sterile 15 ml Conical tube. Following incubation on ice for 20 min, the sample was centrifuged at 4 °C for 5 min at $500 \times g$. The pellet was resuspended in 2 ml of new 0.1X lysis buffer solution and incubated on ice for a further 20 min. The sample was filtered using a 40-micron cell strainer and was centrifuged again at 4 °C for 5 min at $500 \times g$. A wash buffer was formulated by combining 3.5 ml of nuclease-free water, 40 µl of Tris-HCl, 8 µl of NaCl, 12 µl of MgCl2, 400 µl of BSA and 40 µl of Tween-20. The cell pellet was resuspended in 1 ml of this wash buffer, before being filtering with a 40-micron cell strainer and transferred into a sterile 1.5 ml Eppendorf tube. Individual nuclei within the solution were counted and viability assessed using the Countess II FL automated cell counter (Thermo Fisher Scientific). Diluted nuclei buffer (10X Genomics Single-Cell ATAC Library and Gel Bead Kit; 1000175) was added to achieve a concentration of 2000 nuclei per µl, with a targeted nuclei recovery of 4000 nuclei.

### scATAC library preparation
Following nuclei extraction, single-nuclei solutions were processed by the 10X Genomics Sequencing laboratory at Genomics Birmingham. The Chromium Single-Cell ATAC Solution and the Chromium Next GEM

Single-Cell ATAC Library and Gel Bead Kit (10X Genomics) were utilised according to the manufacturer's instructions. Briefly, diluted nuclei were added to a transposition solution, where transposase entered nuclei and fragmented the DNA in open chromatin regions, while adaptor sequences were merged onto the DNA fragments. Individual transposed nuclei were joined with a barcoded gel bead and partitioning oil to create a Gel Bead-in-Emulsion (GEM). By dissolving gel beads, primers and 10x barcodes were added to single-stranded DNA, and the GEM's fragmented to collect the DNA. All transposed DNA fractions from the same nucleus were tagged with individual barcodes to identify the nucleus of origin after DNA pooling.

The samples were cleaned using Solid Phase Reversible Immobilization (SPRI) and silane magnetic beads. DNA was then amplified using PCR and the quality assessed with Agilent TapeStation High Sensitivity D1000 ScreenTape. Sequencing was performed with Illumina NextSeq 500, aiming for 4000 nuclei at a depth of 25,000 read pairs per nucleus.

### scATAC data pre-processing and integration
Reads were processed into FASTQ format and peak-barcode counts using Cell Ranger ATAC v2.0.0, aligning to the prebuilt Cell Ranger human reference genome GRCh38-2020-A-2.0.0. The counts matrix for each sample was read into R and stored as Seurat objects for downstream analysis with Seurat v4.1 and Signac v1.6.0 R packages. For integration of samplewise data, a common peak set was created from overlapping peaks across all samples and filtered to remove those with peak width <20 or >10,000. For each sample, per-cell fragments aligned to this common peak set were calculated using FeatureMatrix() and stored in a new chromatin assay with CreateChromatinAssay(). Normalisation and dimensionality reduction was performed using Signac with latent semantic indexing (LSI) which consisted of running functions RunTFIDF; FindTopFeatures; RunSVD and RunUMAP. This normalises using term frequency-inverse document frequency (TF-IDF) and singular value decomposition followed by UMAP on the top 50% of peaks in terms of their sample variation. LSI components 2:30 were used in UMAP dimensionality reduction. Sample integration was performed using Harmony R package taking the merged sample data using reduction = 'lsi' and the combined peaks assay as input parameters. Sample identifiers were used as the covariate upon harmony integration. Harmony integration aligned cell type across samples as visualised on UMAP dimensionality reduction (Fig. S11). Gene activity scores are calculated for each gene in the genome by summing the fragments intersecting the gene body and promotor region. Gene co-ordinates are extracted and extended 2kbp upstream to include the promotor region.

### Unsupervised clustering and major cell lineage annotation
Harmony-adjusted cell embeddings were used for unsupervised clustering using Seurat functions FindNeighbours() on harmony dimensions 2:30 to generate the shared neighbour graph and FindClusters() to identify clusters, optimising the modularity with the Louvain algorithm. Resolution parameter was set to 0.2 for coarse-grained clustering. To estimate per-cell gene activity from peaks data, the Signac function GeneActivity() was used, and cluster marker peaks and genes were identified using FindAllMarkers() on the combined peaks and gene activity assays respectively. Clusters were annotated with major cell type lineage Lymphocyte, Myeloid, Fibroblast and Epithelial based on canonical cell type marker gene activity profile and chromatin accessibility profile at known major cell type defining genes.

### Differential peak and gene activity analysis
Peak-level differential analysis identified peak sites with differential chromatin accessibility between major cell lineages and between treatment-exposed and treatment naïve tissue using FindMarkers() on the combined peaks assay. The statistical test used a logistic regression framework to determine differentially accessible peaks by constructing a logistic regression model predicting group membership based on each feature individually and compares this to a null model with a likelihood ratio test. Latent variables were set to the peak region fragments. Peaks were considered differentially accessible if absolute log2FC > 0.25 and Benjamini–Hochberg adjusted $p$

value < 0.01. For gene activity differential analysis FindMarkers() was used with a minimum percentage of cells set to 0.1 and logFC threshold of 0.2. Gene activity was regarded as differential if Benjamini–Hochberg adjusted *p* value < 0.001 and absolute log2FC > 1.

## Differential transcription factor motif enrichment

Transcription factor motif position frequency matrices were obtained using the R package JASPAR2020 and motif information added to the Signac object using AddMotifs(). Per cell motif activity scores were calculated with RunChromVar() from the R package chromVar[22] on BSgenome.Hsapiens.hg38 and the combined peaks assay. Differential activity of motifs was assessed between major lineage cell types and between treatment-naive vs. post-chemotherapy tissues using FindMarkers() on the 'chromvar' motif activities with fold change calculation computing the average difference in z-score between groups.

## Copy number alterations

Genome-wide copy number alterations (CNAs) for individual cells were calculated using the epiAneufinder algorithm[57]. This uses the read counts from scATAC-seq data as a proxy for the number of DNA copies in segmented genomic regions. To overcome sparsity, lowly covered cells were filtered out and the genomic bin size was set at 100,000 bp. To avoid bias, ENCODE blacklisted regions and genomic bins with zero counts in >85% of cells were removed.

## Signature scoring

Cells were scored for signature gene sets of interest using Seurat AddModuleScore() on gene activity. This score is calculated as the average activity of the gene set per single cell minus background expression from randomly selected control features with positive scores indicating that the gene module is expressed more highly than expected given the average population expression. Published gene sets of interest were retrieved from MSigDB. The HGSOC tumour stress signature gene set was retrieved from previous study[10].

## Immunohistochemistry staining

Sections of formalin fixed paraffin embedded tissue were stained using a Leica Bond 3 automated immunostainer (Leica, Bannockburn, IL) using antibodies obtained from Leica (catalogue ref. NCL-L-p53-D07). A high pH epitope retrieval buffer was used before the application of primary antibodies. Detection was performed using Leica Bond-detection kits (catalogue ref. D59800), followed by hematoxylin counterstains.

## Statistics and reproducibility

Wilcoxon rank sum test was applied to compare the distributions of cell type proportions and signature scores stratified by conditions of interest. All correlation analysis were conducted using Spearman Rank test. Sample size ($n = 10$) consisted of 5 treatment-naive and 5 NACT-treated samples with one matched pre/post treatment pair being derived from the same patient (patient A).

## Reporting summary

Further information on research design is available in the Nature Portfolio Reporting Summary linked to this article.

## Data availability

Sequencing data have been deposited in Gene Accession Omnibus with accession number GSE247982.

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

## Acknowledgements

This work was supported by Wellbeing of Women (WoW) grant reference ELS810, 'The use of a novel single-cell sequencing technology, Drop-Seq, to unravel tumour heterogeneity and identify potential chemo-sistant cells in high-grade serous ovarian cancer (HGSOvCa)' and by Medical Research Council (MRC) MR/S02235X/1, 'Interrogating the relationship between tumour microenvironment and its infiltrating immune cells in high-grade serous ovarian cancer'. The work was also supported by OVACOME 'A temporal sequence assessment of epigenetic and gene mutational alterations in patients diagnosed with ovarian carcinomas'. This is independent research funded by MRC and Ovacome and carried out at the National Institute for Health and Care Research (NIHR) Birmingham Biomedical Research Centre (BRC). The views expressed are those of the author(s) and not necessarily those of the MRC, Ovacome or the NIHR or the Department of Health and Social Care.

## Author contributions

Conceptualisation: P.M., J.Y., S.K., D.J., K.S., J.B., S.S., and S.O. Methodology (sample and data acquisition): R.P., A.W., and R.G. Methodology (data analysis): W.C. Investigation: W.C. and R.P. Supervision: P.M., J.Y., and J.Z. Writing (original draft): W.C. Writing (review & editing): R.P., P.M., J.Y., S.O., A.W., J.Z., and S.S.

## Competing interests

The authors declare no competing interests.
