## [Peer Review File · Communications Biology]

This manuscript has been previously reviewed at another *Nature Portfolio* journal. This document only contains reviewer comments and rebuttal letters for versions considered at *Communications Biology*.

Reviewers' comments:

Reviewer #1 (Remarks to the Author):

Croft et al. profiled single cell chromatin accessibility landscape of omental OC metastasis from treatment-naïve and neoadjuvant chemotherapy-treated patients and defined the chromatin accessibility profiles of epithelial, fibroblast, myeloid and lymphoid cells. However, their results look heavily confounded by sample/batch despite authors claimed that they integrated samples using harmony.

Figure 1B, epithelial cluster is clearly driven by samples preB, preD and preG. Which covariate was used for sample integration? The bottom umap colored by clusters show no localized structures with different clusters bleed through each other.

Figure 1D, epithelial cluster is annotated as tumor. However, based on the marker signatures shown, this cluster look more like a mix of both normal and tumor epithelium. Did authors enrich for tumor cells before single cell sequencing? Can authors show the purity of metastasis collected?

Figure 5A, a re-clustering of epithelial cells is needed, as the previous un-supervised clustering is to capture the difference among various cell types. Again, sample specific effects are dominated in this umap and needs to be taken care of.

Reviewer #2 (Remarks to the Author):

Summary

To better understand transcriptional regulators of treatment response, Croft et al apply scATAC-seq to omental metastases of treatment naïve and chemotherapy-treated high-grade serous ovarian tumours. They predicted transcription factors (TFs) associated with malignant cell identity (MEIS/PBX) and examined how the activity of TFs change in chemotherapy-treated cells (gained activity of TP63, reduced cell cycle activity via E2F2/E2F4). They also identified TFs whose predicted activity correlates with an established stress signature linked to chemoresistance. I think this topic is interesting

relevant to the field, and the data itself is valuable resource.

The novelty of the study would be the identification of specific TFs that may underlie the stress response and potentially chemoresistance. The biggest weakness of the study is that, given the design of the study—ie being restricted to analysis of a scATAC-seq cohort—the findings are fairly speculative/correlative. Although the conclusions would benefit from functional validation, I understand that it may be beyond the scope of the study. Despite this, I believe the manuscript reveals several interesting findings that could set the stage for subsequent studies, so I support its publication.

I reviewed this manuscript for a previous journal and commend the authors for addressing my previous comments prior to this submission. As such, my remaining comments are minor.

Comment 1

The authors use predicted “gene activity” (ie. accessibility throughout gene body) to support that the putative transcription factors have increased expression in the post-NACT samples. I wonder if this could be further substantiated by either (or both): 1) showing the ATAC trace throughout the region to help convey that it looks like an actively transcribed region, or 2) use the scRNA-seq data from Zhang et al (Sci Adv, 2022) to look at expression in post-NACT samples to see if expression is higher.

Comment 2

I think the paragraph on study limitations could be expanded somewhat. It is important to mention somewhere that it is critical to functionally evaluate the involvement of these factors. Also, the manuscript builds a narrative on select transcription factors (TP63, TP53, TWIST1, MEIS/PBS) based on relevance to previous literature, but the analyses pulled out many other factors that may be equally likely to underlie resistance.

Comment 3

Throughout the manuscript, “epigenetic regulation” is used. But this is really just looking at transcription factors and binding sites. While sure, there is an epigenetic element (redistribution of nucleosomes), the primary interest seems to be more “gene regulatory mechanisms”. I would argue it makes more sense to say that than “epigenetics”.

Comment 4

Figure 3D says “differential peaks”

Comment 5

Fig S5 - "Epithelial cell proportion was associated with reduced myeloid and fibroblast fraction...". With only 4 cell types, it is necessary that higher proportion of one cell type will decrease the proportion of others.

Reviewers' comments:

Reviewer #1 (Remarks to the Author):

Croft et al. profiled single cell chromatin accessibility landscape of omental OC metastasis from treatment-naïve and neoadjuvant chemotherapy-treated patients and defined the chromatin accessibility profiles of epithelial, fibroblast, myeloid and lymphoid cells. However, their results look heavily confounded by sample/batch despite authors claimed that they integrated samples using harmony.

We thank the reviewer for taking the time to assess our manuscript.

The data is integrated using Harmony (PMID: 31740819) as outlined in the methods section. We have included an additional supplementary figure (Fig. S11) to illustrate this by showing the before and after integration UMAP embeddings and expanded the methods section with details of the integration procedure.

Line 527 edit: “Sample integration was performed using Harmony R package taking the merged sample data using reduction = “lsi” and the combined peaks assay as input parameters. Sample identifiers were used as the covariate upon harmony integration. Harmony integration aligned cell type across samples as visualised on UMAP dimensionality reduction (Fig. S11).”

The integration successfully aligns patients non-tumor cell type populations (sharing broadly similar chromatin accessibility profiles), whilst the heterogeneous tumor populations are less well aligned. This is not unexpected for a tumor characterised by chromosomal instability and is consistent with the extremely high degree of inter and intra-tumor heterogeneity for HGSOC tumor cells (PMID: 37894756; PMID: 37220750).

Additionally, previous single cell transcriptome studies have shown similarly patient-specific tumor transcriptional and copy number variation profiles on post-integrated data (PMID: 35935940; PMID: 35196078).

Supplementary Figure S11. Harmony integration. UMAP embedding of merged (Before Integration) and Harmony-integrated (After Integration) scATAC-seq data overlaid with patient sample label (top) and major lineage cell type (bottom).

Figure 1B, epithelial cluster is clearly driven by samples preB, preD and preG. Which covariate was used for sample integration? The bottom umap colored by clusters show no localized structures with different clusters bleed through each other.

Thank you for this comment. Harmony integration was applied using the lsi embeddings and a unified peak-set with sample identifier as the covariate to integrate data over all samples.

Unbiased clustering is applied using the Louvain algorithm which identifies communities in a network graph structure and is recommended as a “best practise” for clustering single cell data (PMID: 37002403). The majority (8/10) of samples are dominated by epithelial cells and expectedly, there are more epithelial cells from pre-NACT treatment but all samples are represented in reasonable proportions within the total Epithelial cell pool (see Fig.1G).

We applied coarse grained clustering to the harmony-integrated embeddings to group together clusters from the same major lineage cell types. With this clustering, it is primarily the largest differences in accessibility profile being considered for cluster assignment. Finer grained clustering, increasing the resolution parameter would indeed resolve more localised cluster structures. We have included an additional supplementary figure profiling the re-clustered Epithelial subpopulations (Fig. S8)

Figure 1D, epithelial cluster is annotated as tumor. However, based on the marker signatures shown, this cluster look more like a mix of both normal and tumor epithelium. Did authors enrich for tumor cells before single cell sequencing? Can authors show the purity of

metastasis collected?

Thank you for this comment and questions. Figure 1D shows the gene activity profile of Epithelial (EPCAM) and HGSOC tumor (MUC16, WFDC2, LAPTM4B) markers. We use an annotation of “Epithelial” for this cluster (See 1C) as these cells are dominantly marked by EPCAM activity. We can’t absolutely rule out the possibility of some limited ‘normal’ epithelial cell presence but both the tumor marker activity (Fig. 1D) and copy number alteration data (Fig. 1E) evidence the majority of Epithelial cells being tumor. Furthermore, an expert ovarian cancer histopathologist reviewed the sample tissue images and confirmed that no non-neoplastic epithelial structures could be identified.

We have now included additional HGSOC tumor markers (KRT7 and PAX8) and DNA damage activated genes (PARP1, PARP2) to Figure 1D (and Figure 5B) showing broadly positive activity in the Epithelial population. At least one of the DNA damage induced genes; PARP1 or PARP2, show increased activity over all epithelial cells (Fig.1D, Fig.5B), adding to the evidence that these are malignant tumor, rather than normal cells.

Tumor cells were enriched at the point of sampling in that macroscopic tumor-rich regions of tissue were selectively isolated by an experienced surgeon at the time of sampling. We have added further detail to the methods and a supplementary figure added with representative images of the omentum tissue from samples pre_E and post_H (Fig. S9).

Line 460 added: “Macroscopic epithelial tumor cell rich tissue regions were isolated and immediately snap frozen in liquid nitrogen. Biopsies were examined by an Ovarian Cancer expert histopathologist to confirm diagnosis of metastatic HGSOC. Expert also confirmed the presence of neoplastic structures with no evidence of other non-neoplastic epithelial structures within the samples (Fig. S9)”.

Line 5 pathologist co-author added: A. Williams⁴

Supplementary Figure S9. HGSOC Omental metastasis tissue imaging. Representative H&E staining images from a pre NACT tissue sample (pre_E) and two regions from a post NACT tissue sample (post_H). Arrows indicate tumor regions.

Figure 5A, a re-clustering of epithelial cells is needed, as the previous un-supervised clustering is to capture the difference among various cell types. Again, sample specific effects are dominated in this umap and needs to be taken care of.

Thank you, we agree sample-specific effects are apparent in this umap, this is expected given the nature of HGSOC metastasis as a TP53 mutant-driven tumor that has completely lost control of genomic stability. TP53 is mutated in >96% of HGSOC but there exist a range of different mutation hotspots in the TP53 coding genome, diversely effecting TP53 function

(PMID: 34298679) and thus providing a potential source for the evolved tumor-specific chromatin landscapes that are observed.

In terms of taking care of sample specific effects, we consider these are not confounding sources of unwanted variation as they are still evident following harmony integration and are reflective of the true nature of these highly heterogeneous tumours.

However, we fully agree with the reviewer that a re-clustering of the epithelial cells would be beneficial, therefore we have subset, reintegrated and re-clustered just Epithelial cells (Harmony integration with sample identifier as covariate) and added a UMAP embedding with clustering to Fig. 5A plus a supplementary figure showing the peak, gene and motif activity profiles of the tumor sub-clusters (Fig.S8).

Line 261 edited: “UMAP embeddings overlaid with sample label and unsupervised clustering (Fig.5A), HGSOc tumour and DNA damage marker gene activity, and hierarchical clustering on CNAs (Fig. S7) are suggestive of a high degree of intra and inter-tumoral heterogeneity. Profiling clustered subpopulations highlights heterogeneity of gene, peak and motif activity within tumour subpopulations (Fig. S8) All samples had prominent EPCAM, LAPTM4B, KRT7, PAX8 gene activity plus MEIS1 motif enrichment...”

Added to panel A in Figure 5:

Supplementary Figure S8. Profile of HGSOC tumour cell subpopulations. Scaled average gene activity (top), counts at peak region (middle) and motif activity score (bottom) of the top 5 marker features for each epithelial tumour cell subclusters identified by unsupervised clustering.

Reviewer #2 (Remarks to the Author):

Summary

To better understand transcriptional regulators of treatment response, Croft et al apply scATAC-seq to omental metastases of treatment naïve and chemotherapy-treated high-grade serous ovarian tumours. They predicted transcription factors (TFs) associated with malignant cell identity (MEIS/PBX) and examined how the activity of TFs change in chemotherapy-treated cells (gained activity of TP63, reduced cell cycle activity via E2F2/E2F4). They also identified TFs whose predicted activity correlates with an established stress signature linked to chemoresistance. I think this topic is interesting relevant to the field, and the data itself is valuable resource.

We thank the reviewer for taking the time to assess our manuscript and are pleased they feel that it provides interest, relevance and a valuable data resource for the field.

The novelty of the study would be the identification of specific TFs that may underlie the stress response and potentially chemoresistance. The biggest weakness of the study is that, given the design of the study—ie being restricted to analysis of a scATAC-seq cohort—the findings are fairly speculative/correlative. Although the conclusions would benefit from functional validation, I understand that it may be beyond the scope of the study. Despite this, I believe the manuscript reveals several interesting findings that could set the stage for subsequent studies, so I support its publication.

Thank you for the comment, we agree that functional validation was beyond the scope here but as stated, we too hope that the presented analysis results provide interesting avenues highly worthwhile for others in the field to explore and validate.

I reviewed this manuscript for a previous journal and commend the authors for addressing my previous comments prior to this submission. As such, my remaining comments are minor.

Comment 1

The authors use predicted “gene activity” (ie. accessibility throughout gene body) to support that the putative transcription factors have increased expression in the post-NACT samples. I wonder if this could be further substantiated by either (or both): 1) showing the ATAC trace throughout the region to help convey that it looks like an actively transcribed region, or 2) use the scRNA-seq data from Zhang et al (Sci Adv, 2022) to look at expression in post-NACT samples to see if expression is higher.

Thank you for the comment, we have added traces to Figure 3 panel G to support the expression of selected TFs of interest.

Panel G added to figure 3:

Fig. 3. Modulation of the chromatin accessibility landscape and transcription factor binding motif enrichments following chemotherapy. **A** Proportion of total stratified by chemotherapy. Points represent the within-sample cluster fraction and p -value determined by Mann-Whitney test. **B** Summary counts of differentially accessible chromatin (DAC) sites open/closed post chemotherapy. **C** Differential accessibility of chromatin sites between

treatment-naïve (pre) and post chemotherapy (post) samples. Coloured points indicate DACs (adjusted $p < 0.01$ and absolute $\log_2FC > 0.25$). Motif sequence logos presented for the top 3 motifs enriched within open/closed peak sites. **D** Aggregated accessibility profile of selected DACs stratified by treatment. Tracks Y scale represents normalized fragment count. **E** Within patient-cell-type average transcription factor motif chromVar activity score profile of motifs identified as differentially enriched (adjusted $p < 0.01$). **F** Intersections of differentially increased gene activity (expression) with enriched motifs in pre NACT treated tumour cells (top) and post NACT-treated tumour cells (bottom). **G** Aggregated accessibility profile of selected transcription factors.

Line 220 edit: “Expression of these TFs in tumour cells is supported by accessibility in the gene body/promoter (Fig.3G) and they represent candidate important factors shaping the post-chemotherapy transcriptional landscape of tumour cells.”

Comment 2

I think the paragraph on study limitations could be expanded somewhat. It is important to mention somewhere that it is critical to functionally evaluate the involvement of these factors. Also, the manuscript builds a narrative on select transcription factors (TP63, TP53, TWIST1, MEIS/PBS) based on relevance to previous literature, but the analyses pulled out many other factors that may be equally likely to underlie resistance.

Thank you for the suggestion, we have expanded the limitations section to include these very valid points.

Line 439 edit: “Data presented here highlights multiple transcription factors with high likelihood to be of importance in shaping tumor cell resistance but further work to functionally evaluate these factors is required.”

Comment 3

Throughout the manuscript, “epigenetic regulation” is used. But this is really just looking at transcription factors and binding sites. While sure, there is an epigenetic element (redistribution of nucleosomes), the primary interest seems to be more “gene regulatory mechanisms”. I would argue it makes more sense to say that than “epigenetics”.

Thank you for the suggestion, chromatin accessibility is of course a layer of epigenetic regulation influencing gene expression, but we have toned down the use of the more generic term “epigenetic regulation” and replaced with “gene regulatory” where appropriate.

Line 48 edit: “...gene regulatory landscape...”

Line 78 edit: “Less well understood are the gene regulatory mechanisms orchestrating transcriptional programs...”

Line 80 edit: “...important gene regulatory control mechanism...”

Line 428 edit: “...impact of NACT on the gene regulatory landscape of...”

Line 448 edit: “...insights into the gene regulatory mechanisms of...”

Comment 4

Figure 3D says “differential peals”

Thank you, this has now been corrected to “differential peaks”

Comment 5

Fig S5 - "Epithelial cell proportion was associated with reduced myeloid and fibroblast fraction...". With only 4 cell types, it is necessary that higher proportion of one cell type will decrease the proportion of others.

Thank you. We agree that the cell type compositions could be inherently linked when interrogating a small number of cell types and this may not represent an ‘association’. We have edited this sentence for clarity.

Line 133 edit: “Samples with increased epithelial proportion had concurrently reduced myeloid and fibroblast fraction and trended towards shorter survival times (Fig. S5).”

REVIEWERS' COMMENTS:

Reviewer #1 (Remarks to the Author):

My concerns are properly addressed. I support its publication.

Reviewer #2 (Remarks to the Author):

My initial comments were only minor and I thank the authors for taking the time to address them carefully. I believe they have been addressed. Together, I am satisfied with the revised manuscript and would be happy to see it published.